# Automatic Registration of Optical Images with Airborne LiDAR Point Cloud in Urban Scenes Based on Line-Point Similarity Invariant and Extended Collinearity Equations

**DOI:** 10.3390/s19051086

**Published:** 2019-03-03

**Authors:** Shubiao Peng, Hongchao Ma, Liang Zhang

**Affiliations:** 1School of Remote Sensing and Information Engineering, Wuhan University, Wuhan 430079, China; shbpeng@whu.edu.cn; 2Jiangsu Surveying and Mapping Engineering Institute, Nanjing 210013, China; 3Faculty of Resources and Environmental Science, Hubei University, Wuhan 430062, China; zhangliang_hubeiu@hotmail.com

**Keywords:** registration, LiDAR point cloud, point-line similarity invariant, line matching, extended collinearity equations (ECE)

## Abstract

This paper proposes a novel method to achieve the automatic registration of optical images and Light Detection and Ranging (LiDAR) points in urban areas. The whole procedure, which adopts a coarse-to-precise registration strategy, can be summarized as follows: Coarse registration is performed through a conventional point-feature-based method. The points needed can be extracted from both datasets through a matured point extractor, such as the Forster operator, followed by the extraction of straight lines. Considering that lines are mainly from building roof edges in urban scenes, and being aware of their inaccuracy when extracted from an irregularly spaced point cloud, an “infinitesimal feature analysis method” fully utilizing LiDAR scanning characteristics is proposed to refine edge lines. Points which are matched between the image and LiDAR data are then applied as guidance to search for matched lines via the line-point similarity invariant. Finally, a transformation function based on extended collinearity equations is applied to achieve precise registration. The experimental results show that the proposed method outperforms the conventional ones in terms of the registration accuracy and automation level.

## 1. Introduction

High spatial resolution optical images acquired by aerial or satellite remote sensors are one of the most commonly used data sources for geographic information applications [1]. They have been used for the detection and extraction of manmade objects, urban planning, environmental monitoring, rapid responses to natural disasters, and many other applications [2,3,4,5]. Nonetheless, the lack of three-dimensional (3D) information in optical images limits their applications in 3D scenes [6,7,8]. Airborne Light Detection and Ranging (LiDAR), on the other hand, has the ability to directly acquire 3D geo-spatial data of the Earth’s surfaces under the World Geodetic System (WGS84). A LiDAR system can operate under a wide range of weather conditions with the acquired dataset free of shadow. Additionally, a laser pulse can penetrate the gaps of plant foliage and hit the ground; hence, not only providing an efficient way for high accuracy Digital Elevation Model (DEM) acquisition in regions covered with vegetation, but also being an indispensable means for forestry parameters retrieval. Both spatial and spectral information can be acquired when optical images and LiDAR point clouds are combined, which effectively compensates for the deficiency caused by a single data source; this combination has many potential applications in natural hazard assessment [9,10,11], true orthophoto production [12,13], change detection [14,15,16], and automatic manmade objects extraction and modeling [17,18,19,20,21], to mention only a few.

The fusion of LiDAR data and optical images can only be performed if they are precisely registered in order to eliminate the geometric inconsistency between the two datasets [22]. Although georeferenced aerial images and airborne LiDAR data should have been precisely registered, misalignments may exist because of the systematic errors of their respective sensor systems [23]. For example, errors may appear due to the insufficient accuracy of the Global Positioning System (GPS) and Inertial Measurement Unit (IMU) observations, or be caused by inappropriate system calibration. Therefore, precise registration is necessary, even if images have been georeferenced prior to fusing with point clouds.

In digital image processing, image registration refers to the process of aligning two or more images pixel by pixel by using a transformation: one of them is referred to as the master and any others which are registered to the master are termed slaves. Registration is currently conducted with intensity-based methods, feature-based methods, or a combination of the two [24,25]. Intensity-based methods involve maximizing the similarity in intensity values between the transformed slave image and the master. Mutual Information (MI) [23] and normalized gradient fields are two of the most often used similarities [26]. In theory, intensity-based methods are fully automatic, but in practice, they are often mathematically ill-posed, in the sense that the registration solution might not be unique and a small change within the data might result in large variation in registration results [27]. In addition, different types of sensors significantly affect the similarity between images; therefore, the choice of similarity measure for the intensity-based methods is very important. In contrast, feature-based registration starts with feature primitives extraction from the master and slave images, respectively. Features can be points, line segments (edges), or surface patches. Then, algorithms are applied to search for matched primitives that will be employed for the establishment of the transformation model, which can be both parametric and non-parametric [25]. The slave image is then rotated, translated, and scaled, according to the established transformation model, after which the conjugate features are identically aligned.

Unlike the image-to-image registration scenario, the registration between optical images and laser scanning data is characterized by registering continuous 2D image pixels to irregularly spaced 3D point clouds, thereby making it difficult to meet the requirements imposed by traditional methods mainly developed for registering optical images. Studies conducted over the past decade in order to solve this problem fall into the following three categories:

(1) LiDAR data is converted to 2D images according to their elevation and intensity values, and optical images are then registered to them by traditional image-to-image registration methods. Mastin et al. [23] suggested the use of mutual information as a similarity measure when LiDAR point clouds and aerial images were to be registered. Parmehr et al. [24] used the normalized combined mutual information (NCMI) approach as a means to produce a similarity measure that exploits the inherently registered LiDAR intensity and point cloud data, so as to improve the robustness of registration between optical imagery and LiDAR data. Abedini et al. [28] applied the scale-invariant feature transform (SIFT) algorithm for the registration of LiDAR data and photogrammetric images. Palenichka et al. [29] utilized salient image disks (SIDs) to extract control points for the registration of LiDAR data and optical satellite images. Experimental results have proved that the SIDs method is more effective than the other techniques for natural scenes. However, it is too complex and computationally expensive to implement in real world applications.

Image-based methods make full use of existing algorithms for image registration and are therefore the methods typically used for the registration of point clouds and optical images. However, due to the inevitable errors caused by the conversion of irregularly spaced laser scanning points to digital images (an interpolation process), and the mismatching present between these two datasets, the registration accuracy may not be as satisfactory as would be expected.

(2) Dense photogrammetric points are first extracted by stereo-image matching, and 3D to 3D point cloud registration algorithms, such as Iterative Closest Point (ICP) or Structure from Motion (SFM), are then applied to establish a mathematical model for transformation [30,31,32]. The main drawback of this method is that the registration accuracy may be influenced by the result of image matching. The matching results may be degraded if not enough salient features can be extracted in regions of shadows, forests, desert, seashores, etc. Moreover, methods in this category require stereo images covering the same area as covered by point clouds, which increases the cost of data acquisition.

(3) Some researchers have tried to establish a direct transformation function between the two datasets. Habib et al. used planar features [33] to establish a direct-mapping model between optical remote sensing images and LiDAR point clouds. Zhang et al. [34] performed the registration of aerial imagery and point clouds by utilizing the inherent geometrical constraint of the object boundaries. Ding et al. [35] made use of the vanishing points to extract features named 2DOCs (2D orthogonal corners) for refining the camera parameters referring to the point clouds. Wang and Neumann [36] further proposed a novel feature called 3CS (3 connected segments) to develop a robust automatic registration approach, which is claimed to be more distinctive than 2DOCs.

Methods in categories (1) and (2) include the indirect registration between optical images and LiDAR point clouds, because both of them require an intermediate step to either convert irregularly spaced point clouds to raster structured images, or to generate photogrammetric point clouds from stereo images. Such intermediate processing can cause errors that are sometimes too serious to be neglected. For instance, points may be missing or sparsely distributed in a calm and clean waterbody due to its absorbent characteristics in the infrared band that a laser scanner often adopts, causing errors when these points are converted to image data by interpolation. Such errors can be controlled in some way with the methods in category (3). However, one should bear in mind that the automation of this type of method is limited by the state-of-the-art algorithms of extracting and matching conjugate features, which are far from mature. In addition, the existing transformation functions in literature are too mathematically complex to be implemented numerically without any difficulties [28,30].

This paper proposes a novel method for automatic registration of the two datasets acquired in urban scenes by using a direct transformation function based on collinearity equations and point features as control information, which falls into method category (3). The proposed method follows a coarse-to-precise registration strategy in which optical images are taken as slaves, and the LiDAR point cloud as the master. If the optical images have not been geo-referenced, then they will be coarsely registered by the conventional image-to-image registration method at first, in which Direct Linear Transformation (DLT) [37] is applied as the transformation function with point features extracted by a Förstner detection [38] as the control information for the establishment of the DLT. Coarse registration is skipped if images have been geo-referenced. In the precise registration stage, extended collinear equations (ECE), which map points in the object world onto their corresponding pixels in the image space, are adopted as the transformation function. Conjugate points extracted from the point cloud and image, respectively, are applied as the control information to calculate the unknown parameters of ECE, which are mainly the six exterior elements of the camera. Unlike existing studies which extracted conjugate points by various point detectors, in our proposed method, they are from the point features detected in the coarse registration stage, and located by matched lines on which the conjugate points lie, thereby overcoming the difficulty of extracting salient points from irregularly spaced LiDAR data. Automatic line matching is based on the line-point similarity invariant–an algorithm which will be described in Section 3. The whole procedure is shown in Figure 1.

The main contributions of this paper are twofold: direct registration by extended collinear equations and automatic matching of line features through the line-point similarity invariant. The first contribution allows for establishing direct correspondence between conjugated line features, other than using these features to estimate the exterior orientation elements of the camera acquiring the optical image and then georeference it. In the context of the registration of the point cloud and optical images, some specific research works were conducted in terms of exterior orientation with the help of point clouds. As a matter of fact, many research works conducted exterior orientation elements estimation from line or plane features. Zhang et al. [39] used building corners as registration primitives and exterior orientation parameters were refined by bundle adjustment using the corner points or centroids of plane roof surfaces as control information. Zheng et al. [40] and Huang et al. [41] used tie points after bundle adjustment of the sequence images to register with the laser point cloud by the ICP method and the homologous laser points acquired by registration were used to optimize the interior and exterior orientation elements of the optical image. Habib et al. [42] described an alternative methodology for the determination of exterior orientation for optical images, which uses three-dimensional straight-line features instead of points. Mastin et al. [23] proposed a novel application of mutual information registration methods, which exploits the statistical dependency in urban scenes of optical appearance with measured LiDAR elevation, to infer camera pose parameters. Yang et al. [43] proposed a method for estimating the exterior orientation parameters of images in the LiDAR reference frame with the conjugate building outlines by extracting them from laser scanning points and optical images, respectively, with the help of direct geo-referencing data. Sungwoong et al. [44] and Armenakis et al. [45] established the relationships between planes in image space and the LiDAR point cloud, respectively, for optimizing the exterior orientation elements of optical images. However, the accuracy of the calculated exterior orientation elements by these methods is affected by the accuracy of the extracted line and plane features. Moreover, it is computationally expensive regarding feature extraction from photogrammetric images and LiDAR data, as well as the establishment of the correspondence between conjugate primitives. In our proposed approach, the exterior orientation was encapsulated into the extended collinear equations in some sense. The basic idea behind the second contribution is to use cheaply obtainable matched points to boost line matching via line–point invariants, even if the matched points are susceptible to severe outlier contamination because of the huge difference between LiDAR-based images and optical images. Therefore, our registration strategy not only establishes a direct transformation model which relieves the computational complexity and simplifies the entire registration process, but also exploits point feature detection and localization by the line-point similarity invariant.

In the following sections, emphases are placed on the establishment of the extended collinearity equations-based transformation function and the matching of line features by the line-point similarity invariant. The structure of the paper is organized as follows: Section 2 expounds the details of the transformation function; Section 3 describes the extraction of line features from both the images and point cloud, as well as automatic line matching based on the line-point similarity invariant; Section 4 presents experimental results and discussions; and this paper ends in Section 5 with conclusions.

## 2. Transformation Function Based on the Extended Collinearity Equations

In photogrammetry, collinearity equations describe the relationship between the coordinates of a 2D image pixel (denoted by *α*) with respect to the image coordinate system and its corresponding object point (denoted by *β*) coordinates with respect to an object space coordinate system. The following is the mathematical expression of the equations:(1)$$\begin{array}{c} x_{\alpha }=x_{0}-f\frac{a_{11}\left(X_{\beta }-X_{o}\right)+a_{21}\left(Y_{\beta }-Y_{o}\right)+a_{31}\left(Z_{\beta }-Z_{o}\right)}{a_{13}\left(X_{\beta }-X_{o}\right)+a_{23}\left(Y_{\beta }-Y_{o}\right)+a_{33}\left(Z_{\beta }-Z_{o}\right)}\\ y_{\alpha }=y_{0}-f\frac{a_{12}\left(X_{\beta }-X_{o}\right)+a_{22}\left(Y_{\beta }-Y_{o}\right)+a_{32}\left(Z_{\beta }-Z_{o}\right)}{a_{13}\left(X_{\beta }-X_{o}\right)+a_{23}\left(Y_{\beta }-Y_{o}\right)+a_{33}\left(Z_{\beta }-Z_{o}\right)} \end{array}$$
where x and y are image coordinates of an image pixel and $X_{\beta }$, $Y_{\beta }$, $Z_{\beta }$ are their corresponding object coordinates; $x_{0}$ and $y_{0}$ denote the coordinates of the principal point of the camera acquiring the image; f is the camera’s focal length; $X_{o}$, $Y_{o}$ and $Z_{o}$ are the object coordinates of the camera station at the time when the image was recorded; and $a_{11}$, …, $a_{33}$ are the elements of a rotation matrix which describes the three-dimensional attitude, or orientation, of the image with respect to the object coordinate system. They are calculated from three rotation angles of the camera with respect to the object space coordinate system, usually denoted by φ, ω, and κ [46].

Since LiDAR data can be viewed as object points, while image pixels are their corresponding image points, collinearity equations are in fact the mathematical model which describes the mapping relationship between the two datasets. Such a model can be employed as the transformation function for registration if conjugate points in the image, and in the LiDAR data, can be extracted and matched. However, when the point cloud is irregularly spaced, it is very difficult, if not impossible, to extract salient points from it, which leads to the difficulty of applying the collinearity equations for registration without modification.

To overcome the abovementioned difficulty, we propose a strategy where point features are replaced by line segments. The fundamental idea is as follows: denote two points on a line p in LiDAR data by A and B. If the coordinate values of A and B are known, then the line passing through them can be expressed by the following parametric equation:(2)$$\left[\begin{array}{c} X_{\beta }\\ Y_{\beta }\\ Z_{\beta } \end{array}\right]=\left[\begin{array}{c} X_{A}\\ Y_{A}\\ Z_{A} \end{array}\right]+\lambda_{P}\left[\begin{array}{c} X_{B}-X_{A}\\ Y_{B}-Y_{A}\\ Z_{B}-Z_{A} \end{array}\right]$$
where $\lambda_{P}$ is a scalar that spans from negative infinity to positive infinity and $X_{\beta }$, $Y_{\beta }$, and $Z_{\beta }$ are the coordinates of a point β on the line p. Given A and B, then β traces out the line as β goes from -∞ to +∞. Substituting Equation (2) into (1), the resultants β below are termed as extended collinearity Equations (3):(3)$$\begin{array}{c} x_{\alpha }=x_{0}-f\frac{a_{11}\left(X_{A}+\lambda_{p}\left(X_{B}-X_{A}\right)-X_{o}\right)+a_{21}\left(Y_{A}+\lambda_{p}\left(Y_{B}-Y_{A}\right)-Y_{o}\right)+a_{31}\left(Z_{A}+\lambda_{p}\left(Z_{B}-Z_{A}\right)-Z_{o}\right)}{a_{13}\left(X_{A}+\lambda_{p}\left(X_{B}-X_{A}\right)-X_{o}\right)+a_{23}\left(Y_{A}+\lambda_{p}\left(Y_{B}-Y_{A}\right)-Y_{o}\right)+a_{33}\left(Z_{A}+\lambda_{p}\left(Z_{B}-Z_{A}\right)-Z_{o}\right)}\;\\ y_{\beta }=y_{0}-f\frac{a_{12}\left(X_{A}+\lambda_{p}\left(X_{B}-X_{A}\right)-X_{o}\right)+a_{22}\left(Y_{A}+\lambda_{p}\left(Y_{B}-Y_{A}\right)-Y_{o}\right)+a_{32}\left(Z_{A}+\lambda_{p}\left(Z_{B}-Z_{A}\right)-Z_{o}\right)}{a_{13}\left(X_{A}+\lambda_{p}\left(X_{B}-X_{A}\right)-X_{o}\right)+a_{23}\left(Y_{A}+\lambda_{p}\left(Y_{B}-Y_{A}\right)-Y_{o}\right)+a_{33}\left(Z_{A}+\lambda_{p}\left(Z_{B}-Z_{A}\right)-Z_{o}\right)} \end{array}$$

Supposing that interior elements of the camera are known, and that systematic errors have been removed, normal equations can be established by linearization using first order Tylor expansion of Equations (4):(4)$$\begin{array}{c} v_{x}=\left(x_{\alpha }\right)-x_{\alpha }+A_{11}\nabla X_{o}+A_{12}\nabla Y_{o}+A_{13}\nabla Z_{o}+A_{14}\nabla \phi +A_{15}\nabla \omega +A_{16}\nabla \kappa +B_{11}\nabla \lambda \;\\ v_{y}=\left(y_{\beta }\right)-y_{\beta }+A_{21}\nabla X_{o}+A_{22}\nabla Y_{o}+A_{23}\nabla Z_{o}+A_{24}\nabla \phi +A_{25}\nabla \omega +A_{26}\nabla \kappa +B_{21}\nabla \lambda \end{array}$$
where ($x_{\alpha }$) and ($y_{\alpha }$) denote the calculated image coordinate of point α from Equation (1) given the initial values of interior and exterior orientation elements and the coordinates of β with respect to the space coordinate system. The calculation of coefficients $A_{11}$–$A_{16}$, $A_{21}$–$A_{26}$ and $B_{11}$, $B_{21}$ can be found in [47].

In the case of n pairs of straight lines being matched, the normal Equation (4) can be written as:(5)$$V=[A_{2n\times 6},B_{2n\times n}]\left[\begin{array}{c} t_{6\times 1}\\ \lambda_{n\times 1} \end{array}\right]-L_{2n\times 1}$$
where ***A*** and ***B*** are the matrices of the coefficients $A_{11}$–$A_{16}$, $A_{21}$–$A_{26}$ and $B_{11}$, $B_{21}$, respectively; ***t*** represents the unknown corrections to the initial approximations of six exterior elements; λ is the vector of unknown parameters $\lambda_{p}$ corresponding to different lines; ***L*** is the vector of constants calculated from the collinear equations using approximated initial values of exterior elements; and ***V*** is the vector of residual errors.

The numerical implementation stage begins after straight line segments are extracted from both datasets. Each straight line in the point cloud is expressed by its parametric equation, and line matching is performed. Matched point pairs are then extracted from the matched line pairs; this is achieved through the following process (shown in Figure 2): a point α is selected randomly from a matched line segment *L*’ in the image; then its corresponding point β in the LiDAR data must lie on the line segment matched with *L*’ in the point cloud, denoted by *L*, or its extended part. The accurate position of the point β is determined by fixing the parameter λ, which is achieved by solving normal Equation (5) by least mean squares iteratively, a process known as space resection in photogrammetry. It is obvious that this method relies heavily on accurate line matching, a process which will be described in detail in Section 3.

We will conclude this section with some comments on the extended collinearity equations. If the optical image is acquired by an aerial photogrammetric camera with a positioning and orientation system (POS) as its direct geo-referencing device, then exterior orientation elements obtained by POS can be used as initial values in the iterative least mean squares adjustment process. If, on the other hand, there are no direct geo-referencing devices applied during image acquisition, or if the optical image is acquired by satellites whose ephemeris parameters are not provided, then other transformation functions, such as polynomial functions, can also be extended by the process stated above.

## 3. The Extraction of Registration Primitives and Automatic Matching Based on Line-Point Similarity Invariant

As mentioned above, our proposed method requires line matching. However, there are few existing algorithms for that purpose in existing literature. Bay et al. [47] proposed matching lines on the basis of their appearance and topological layout. Wang et al. [48] proposed a so-called mean–standard deviation line descriptor (MSLD) for matching line segments automatically using just their neighborhood appearance, without resorting to any other constraints or prior knowledge. Lourakis et al. [49] utilized two lines and two points to build a projective invariant for matching planar surfaces with points and lines. Fan et al. [50] proposed matching lines by line–point invariants between a line and its neighboring points. Two kinds of line–point invariants have been introduced in the literature; one is an affine invariant derived from a line and two points, and the other is a projective invariant derived from a line and four points. In summary, most of the existing line matching methods employ textural features of the local area, or relationships between lines, such as distances, ratios of distances, differential seat angles, invariants, etc. Great differences are witnessed in terms of the textural and spectral information of LiDAR data and optical images, since they are acquired by entirely different sensors. Bearing these things in mind, and with the help of the ideas presented in existing literature [51], the line-point similarity invariant is applied to line matching in this paper.

### 3.1. Line Features Extraction from LiDAR Data and Optical Image

A coarse to precise strategy was applied to extract line features from LiDAR data as follows: Firstly, coarse building roof edge lines were detected by converting the point cloud into an image, and then common line detectors were applied, as stated in the literature [51]. The infinitesimal feature analysis method, which combines the scanning property of an airborne LiDAR system, was then introduced to refine coarse edge lines: when the scanning angle of a laser beam is larger than a given threshold value (Figure 3a), then the echo of the beam should be reflected from a building facade. If an edge line is projected onto the ground, then a vertical plane V which contains the edge line and is perpendicular to the ground can be formed. After slightly moving the plane along its normal direction, an approximate cuboid is formed. The term infinitesimal is used to indicate that the distance the plane moves should be small enough (1.5–2 times the average distance laser points), as shown in Figure 3b. The number of points contained in the cuboid can be easily counted. If the points are more than the necessary number of samples required for fitting a plane by the least mean square (LMS) technique, then a plane can be fitted by LMS, that is, the building facade. The same procedure can be applied to refine the roof plane. The intersection line of the refined roof plane and the fitted building facade is exactly the refined edge line. This procedure is then applied to all edges detected in the first step, finishing the extraction of precise lines. If the width of eaves is greater than 1.5–2 times the average distance of laser points, such as the scenario shown on the right of Figure 3a, we simply leave them out since we require only enough line features for the registration purpose, other than completely extracting precise roof edges; and there should be enough line primitives in urban areas even when some lines are neglected.

Much literature exists concerning line features extraction from images. The Line Segment Detector (LSD) [52] method is employed in this paper. Compared with conventional methods that first apply the Canny edge detector, followed by a Hough transform, the LSD method is believed to be more accurate, yielding a smaller amount of false positive and false negative detections, and requires no parameter tuning.

### 3.2. Line Matching Based on Line-Point Similarity Invariant

As has been stated in the beginning of the present section, line matching based on the line-point similarity invariant is a key step in the proposed registration method. The basic principle of the line-point similarity invariant can be summarized as follows: Points are denoted as $a_{i}$, and lines as $q_{i}$, in a LiDAR-derived intensity image; in the optical image, they are denoted as $b_{i}$ and $p_{i}$, respectively. In this paper, line features are denoted by coefficient vectors. For instance, a line Ax+By+C=0 can be denoted as $q_{i}={\left(A,B,C\right)}^{T}$. Given two sets of line features extracted respectively from the LiDAR intensity image $\left(L_{1}=\left\{q_{1},q_{2}\cdots q_{n}\right\}\right)$ and the coarse-registered optical image $\left(L_{2}=\left\{p_{1},p_{2}\cdots p_{m}\right\}\right)$, and a set of matched points $S=\left\{s_{1},s_{2}\cdots s_{k}\right\},\;\mathrm{where}\;s_{m}=\left\{\left(a_{i},b_{i}\right),0\leq m\leq k\right\}$, which was detected by the Förstner algorithm in the coarse registration stage. Suppose that a line q lies on the LiDAR intensity image, and line p is its corresponding optical image, as shown in Figure 4. After coarse registration, the optical image has been projected on the same coordinate frame as the LiDAR intensity image. The relationship between corresponding lines and points can be expressed by an affine transformation approximation:(6)$$q=H^{-T}p$$
where H is the matrix representing the affine transformation and $H^{-T}$ is the inverse transpose. Now assume that $a_{1}$ and $a_{2}$ lie in the neighborhood of q, and $b_{1}$ and $b_{2}$ lie in the neighborhood of p. We present $a_{1}$, $a_{2}$ and $b_{1}$, $b_{2}$ as homogeneous coordinates ${\hat{a}}_{1},{\hat{a}}_{2},{\hat{b}}_{1},{\hat{b}}_{2}$, allowing them to also satisfy the affine transformation:(7)$${\hat{b}}_{i}=H{\hat{a}}_{i},\left(i=1,2\right)$$

Let: (8)$$\begin{array}{c} D\left(a_{1},a_{2},p\right)=\frac{\left|q^{T}{\hat{a}}_{1}\right|}{\left|q^{T}{\hat{a}}_{2}\right|}\;\\ D\left(b_{1},b_{2},q\right)=\frac{\left|p^{T}{\hat{b}}_{1}\right|}{\left|p^{T}{\hat{b}}_{2}\right|} \end{array}$$

Then, D is the ratio of the respective distance of two points to a line. Substituting Equations (6) and (9) into these new Equations (8), the following equality holds:(9)$$D\left(a_{1},a_{2},p\right)=D\left(b_{1},b_{2},q\right)$$

Equation (9) states that the distance ratio is unchanged after the affine transformation. This property is named the line-point similarity invariant, and it can be utilized for line matching. Armed with the line-point similarity invariant, the whole process for the automatic line matching consists of the following steps:Define a rectangular search region surrounding line q in the LiDAR intensity image, whose length and breadth are determined by 2α·length(q) and 2β·length(q), respectively (Figure 5). Parameters α and β control the size of the rectangle, which can be determined empirically. It is optimal in many cases that the parameters are 1.5 to 2 times the length of line q. One side of the rectangle is parallel to line q. Since the optical image has been coarsely registered to the LiDAR intensity image, a corresponding search region that is approximately the same as this one can be formed in the optical image. Find all matching points within these two search regions from set S. Carry out the same process for each extracted line in the intensity image.Considering that not all matched points in a search region are correct, a similarity measure is defined according to formula (10), after the distance ratios have been calculated with (9), which means that if line p matches with line q, and the point pair $a_{i}$, $b_{i}$ are correctly matched, then Sim(q,p) tends toward 1. The similarity measure of each of the matched points in both search regions is calculated, and lines p and q are labeled as a matched pair when Sim(q,p) approaches 1.
(10)$$Sim\left(q,p\right)=e^{\left\|D\left(a_{i},a_{j},q\right)-D(b_{i},b_{j},p)\right\|}$$Lines matched by step (2) may lack robustness. The right part of Figure 4 demonstrates this more clearly: both line pairs $p^{\prime }$, q and $p^{\prime \prime }$, q meet the requirement of the line-point similarity invariant, but neither $p^{\prime }$ nor $p^{\prime \prime }$ is matched to line q. To overcome this problem, the distance between the two lines is introduced as an auxiliary similarity measure. This definition was illustrated by Figure 6, where A, B are the end points of line p, and the distances from points A and B to the line q are denoted by $d_{1}$ and $d_{2}$, respectively. The distance from q and p is then defined by Equation (11). If D is greater than two times the average distance of laser points, then line p is labeled as not matching with line q.
(11)$$D=\frac{1}{2}$$To speed up the matching process, the following strategy is adopted: considering that a pair of initially-matched lines should be nearly parallel after the coarse registration, because most distortions have been eliminated, we set a threshold of angle tolerance and compare it to the acute angle spanned by a given pair of matched lines. Pairs with a spanned angle larger than the threshold are labeled as mismatched, and are deleted from the candidates waiting for matching. Matching speed is accelerated greatly in this way.Repeat steps (1) to (4) until all lines have been traversed.

## 4. Experimental Results and Discussions

An aerial image and airborne LiDAR point cloud in Changchun, China, were used as test datasets. They were acquired by a Leica ALS70 airborne laser scanning system (Leica Geosystems AG, Heerbrugg, Switzerland). As shown in Figure 7, the area is dense, with buildings and plants. The average point distance of the LiDAR data is 0.7 meters. The vertical and horizontal accuracies are approximately 0.10–0.15 m and 0.2–0.3 m, respectively, which were evaluated by ground control points. Images were acquired by a Leica RCD105 digital frame camera. The size of the aerial image used for the experiment was 5412 × 7216 pixels, with an approximate ground resolution of 0.14 m.

In the first step of the proposed method, coarse registration is performed by DLT. The DLT parameters were calculated by matched points extracted by the Förstner operator from both the image and LiDAR data, followed by the extraction of line segments (Figure 8). It is found that there are plenty of line segments distributed uniformly in the point cloud, and each line is surrounded by a certain number of matched points; thus, the condition of the line-point similarity invariant is satisfied. Though the aerial image has been coarsely registered, some roof surfaces are still obviously shifted from their true positions (shown in the top row of Figure 9), indicating that precise registration is necessary.

The proposed method is then performed for the precise registration, and the results are compared visually with the ones obtained by coarse registration. As shown in Figure 9, the registration accuracy has been greatly improved by the proposed method.

A method for quantitative registration accuracy evaluation was also proposed: line segments were firstly sketched manually both in the point cloud and the optical image, respectively, and line segments in the image were then mapped onto the point cloud by collinearity equations. Formula (11) was finally applied to calculate the distance between a line segment in the point cloud and the corresponding one mapped from the image. A smaller distance indicates a more precise registration. In our experiment, forty-nine such line segments were manually selected. They were independent from the observed ones used for the establishment of the registration model and had different directions. Statistics of maximum error (MAX), mean error (MEAN), and root mean square error (RMSE), of the distances between the forty-nine pairs of line segments were used for the accuracy evaluation. Table 1 shows that after coarse registration, the MAX, MEAN, and RMSE are 1.52 m, 0.61 m, and 0.45 m, respectively. They decrease to 0.52 m, 0.24 m, and 0.13 m, respectively, after precise registration. Considering that the average LiDAR point distance is about 0.7 m, and that the ground resolution of the optical image is about 0.14 m, we conclude that the proposed registration method is effective and precise. Figure 10 shows a 3D perspective view of the colorized LiDAR point clouds in the low-density urban setting of Changchun, generated with the registered image and LiDAR data. Although both the image resolution and point density are low, the high quality of registration is readily ascertained through the buildings, trees, and other objects shown in the scene.

It is worthwhile to mention that in our experiment, two types of images were generated based on the point cloud before the Förstner algorithm was performed to extract point features from them: an intensity image and elevation image. The former used intensity values (normalized to 0–255) recorded by the LiDAR system, while the latter used elevations (also normalized to 0–255) as the digital numbers of a pixel. The mismatch ratio of points in the intensity image and optical image is 40.2%, and it is as high as 75.2% in terms of the elevation image and optical image, indicating that the coarse registration based on point features and DLT could not be accurate enough, no matter whether an intensity image or elevation image was used to extract point features from the point cloud.

## 5. Conclusions

A new registration method based on the line-point similarity invariant and extended collinearity equations is proposed in order to solve the automatic registration problem when LiDAR data and optical images are to be integrated in urban scenes. The main contributions of this paper are summarized as follows: (1) infinitesimal feature analysis which fully utilizes the scanning characteristics of a LiDAR system is proposed to refine roof edge lines, thus solving the problem of inaccurate edge lines extraction from LiDAR data; (2) a rigorous mathematical model for direct 2D to 3D registration is developed based on traditional collinearity equations, achieving direct registration between 2D images and a 3D point cloud; and (3) matched lines can be searched automatically with the guidance of matched points extracted by the Förstner operator, with the help of line-point affine similarity in a local area. Point features and line features are effectively combined in the proposed method, achieving precise registration with a high efficiency.

The accuracy of the proposed registration method mainly relies on the accurate extraction of line segments from the point cloud, because it is the master dataset, and registration primitives used for the establishment of the extended collinearity equations are line segments detected from it. The infinitesimal feature analysis method for line extraction from the point cloud guarantees that the errors of the extracted lines are no greater than the average distance of points in the dataset. Thus, if the resolution of the image to be registered is around the average distance of laser scanning points, then a subpixel registration accuracy is expected.

In mountain areas and suburbs lacking in line segments, curves can serve as candidates that are able to mimic linear features. However, how to extend collinear equations to curves remains a technical problem worthy of further study. In addition, the image resolution and LiDAR point density will affect the accuracy and quantity of extracted features; this, in turn, will influence the precision of the final result. For registration application, determination of the optimal image resolution for a given point density, and vice versa, requires extensive experiments.

## Figures and Tables

**Figure 1 sensors-19-01086-f001:**
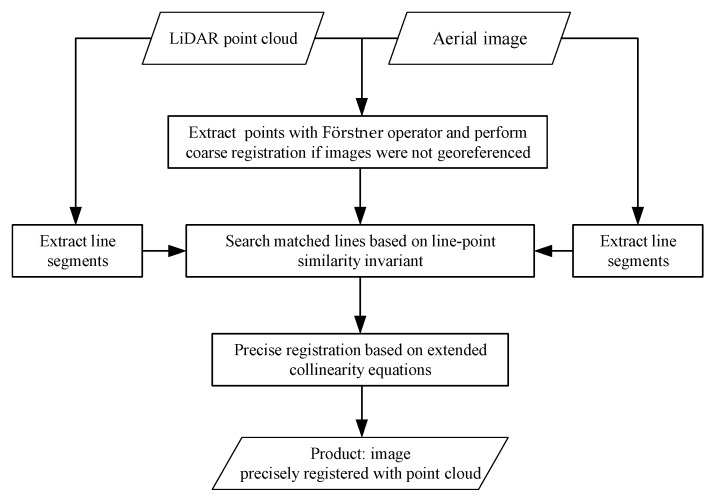
The flow chart of automatic registration of urban aerial images with LiDAR points based on the line-point similarity invariant and extended collinearity equations.

**Figure 2 sensors-19-01086-f002:**
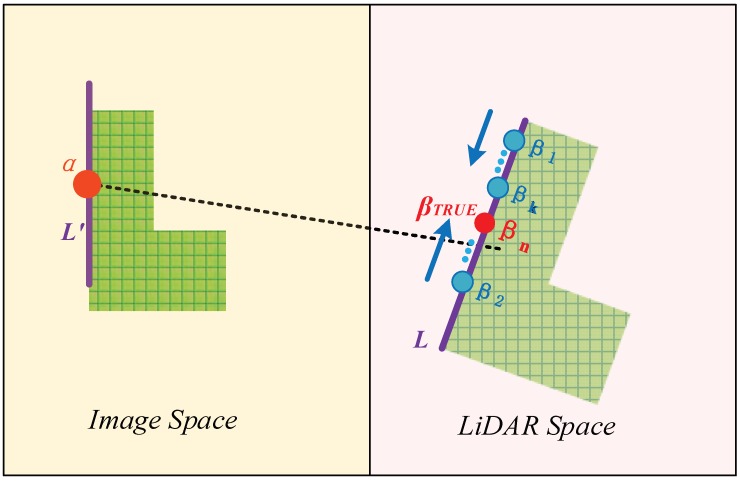
In the iterative process, a point β approaches its true position: *L*’ and *L* constitute a matched line pair lying on image space and LiDAR space, respectively. The true corresponding point is determined as soon as parameter λ is calculated in the iterative process.

**Figure 3 sensors-19-01086-f003:**
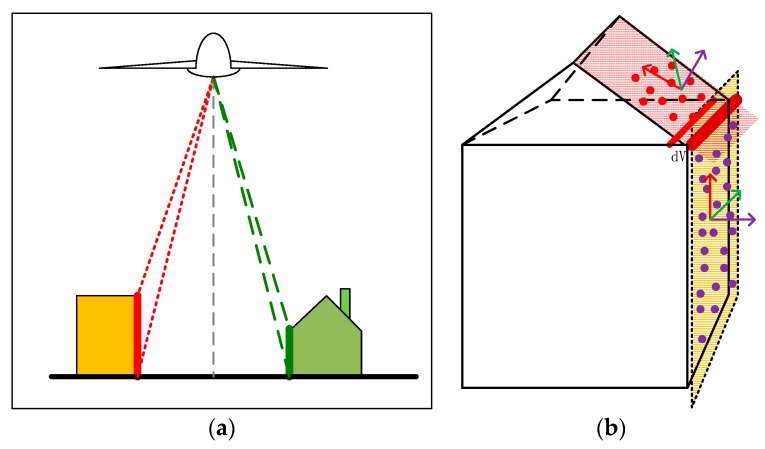
Line features extraction based on LiDAR point clouds: (**a**) LiDAR point clouds echoed from the wall; (**b**) the small cuboid formed by moving the wall or the roof plane along its normal direction.

**Figure 4 sensors-19-01086-f004:**
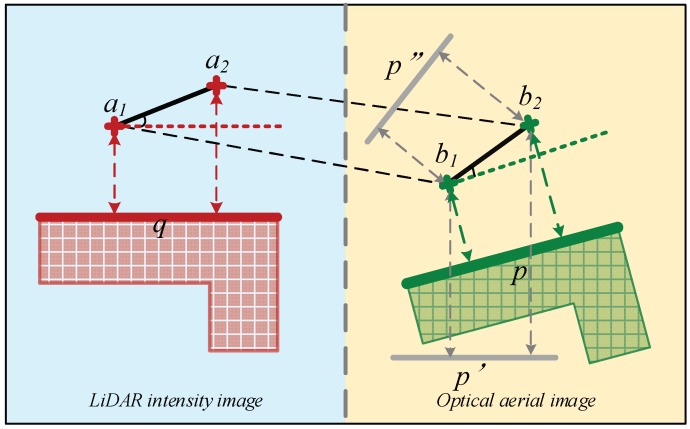
The corresponding relationship between points and lines in a local area.

**Figure 5 sensors-19-01086-f005:**
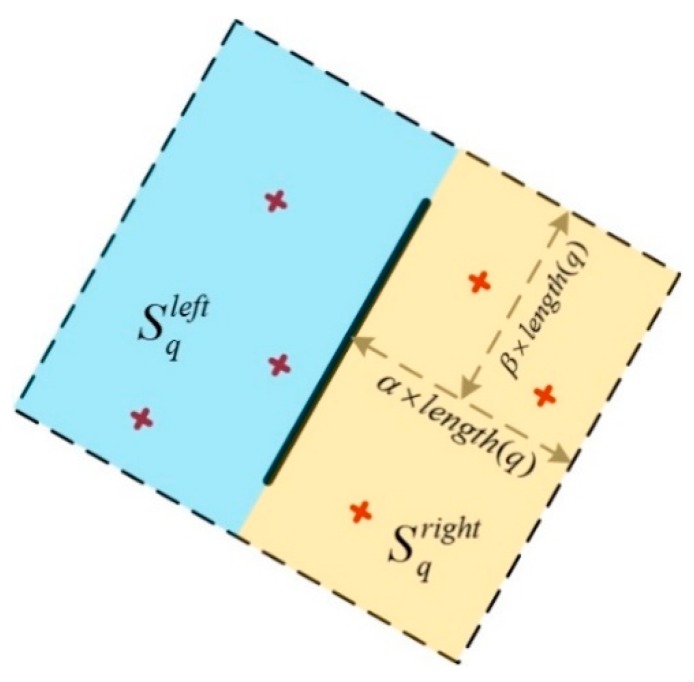
Search region determination.

**Figure 6 sensors-19-01086-f006:**
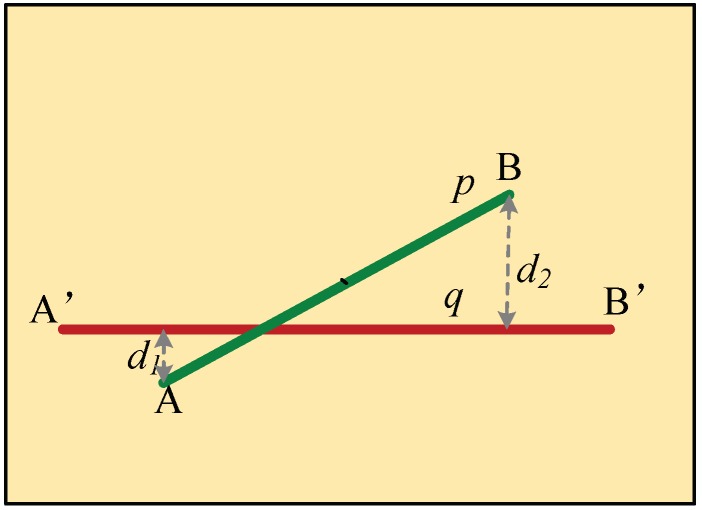
Distance between two straight line segments.

**Figure 7 sensors-19-01086-f007:**
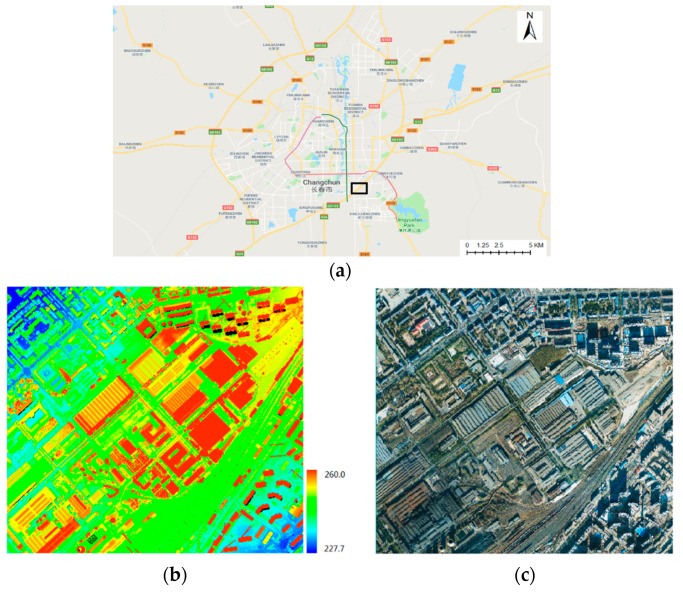
Experimental data: (**a**) location map of experimental area (**b**) LiDAR point clouds and (**c**) aerial optical image.

**Figure 8 sensors-19-01086-f008:**
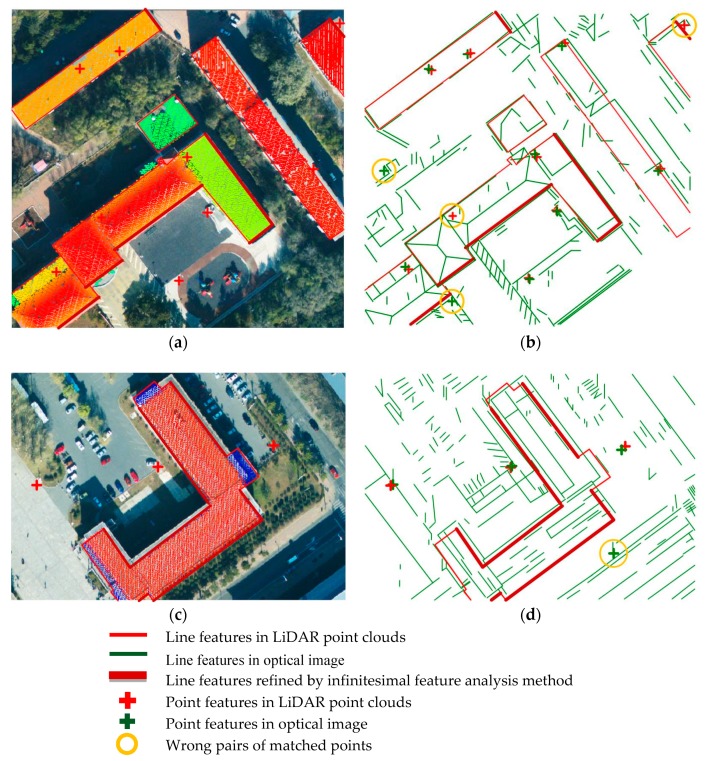
Extracted lines and points: (**a**) and (**b**) show the features extracted from LiDAR data; while (**c**) and (**d**) show the features extracted from optical images.

**Figure 9 sensors-19-01086-f009:**
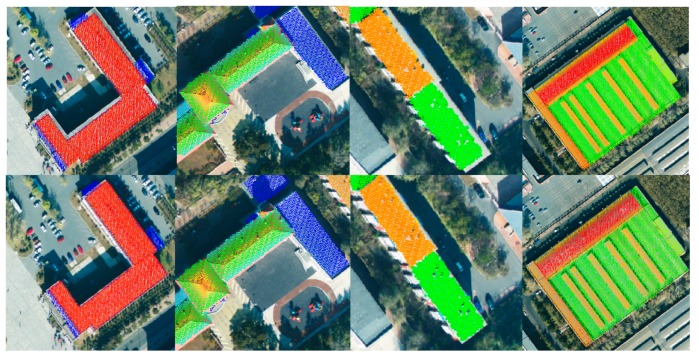
Visual comparison showing the coarse registration results, on the top row, and the precise registration results, on the bottom row.

**Figure 10 sensors-19-01086-f010:**
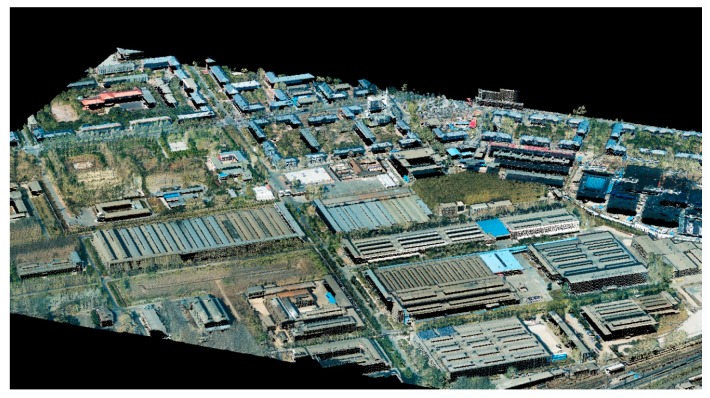
3D visualization of a rural scene in Changchun using the colorized 3D LiDAR point.

**Table 1 sensors-19-01086-t001:** Distances between check pair lines and their statistics.

Registrations	Error Statistics (Unit: Meter)		
	MAX	MEAN	RMSE
Coarse registration by initial exterior orientation elements	2.76	1.36	0.84
Coarse registration by Förstner operator and DLT transformation model	1.52	0.61	0.45
Precise registration by extended collinearity equations	0.52	0.24	0.13

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
