# Peer review of "Automatic Registration of Optical Images with Airborne LiDAR Point Cloud in Urban Scenes Based on Line-Point Similarity Invariant and Extended Collinearity Equations"

_sensors, 2019, doi:10.3390/s19051086_

Round 1
Reviewer 1 Report
Authors provided a nice experiment and methodology-development in the topic automated image registration. The proposed method is convincing, although, there are some limitations in the applicability (to urban scenes). However, I found it well written and inspite of the equation-based mathematical presentation (which was important) easy to read and understand. I have only some minor remarks which can improve the quality:
In paragraph P2L54-70 I am missing the citations
Introduction is well organized and focuses directly on the topic itself, but the aims cab be improved. A better paragraph would be more reasonable for the aims instead of the current descriptive form. Now, it is rather long (L114-139) and readers have to seek a lot the real content. I suggest to point on a research gap of te current studies, focus the interest on it and try to rephrase the aims from a scientific side, maybe with using hypothesis-approach.
Data for experimental results (section 4) needs a bit more detailed information about the input data:
- what was the type of optical sensor?
- what was the density (point/m2) of the LiDAR point cloud?
- please specify the scale unit in Fig. 7, and maybe it would be good to show the place on a draft map
- please provide a legend for Fig. 7/a (now it shows colors without meaning)
Maybe a staistical test can highlight the relevance of the new approach. I am not sure that it is possible but based on the Table 1 it seems that e.g. a paired t-test can be used ...
Author Response
First of all, we would like to thank all the three reviewers for their careful review and valuable suggestions and comments, which inspired us to rethink some statements and rewritten some paragraphs. Attached please find the point-to-point responses to the comments.

Reviewer 2 Report
The authors showed a well designed point and line feature detection and matching scheme in support of the task toward automatic registration of optical images with airborne LiDAR point cloud in urban scenes. The rich appearances of straight line segment in urban environment makes this application possible. The methodology proposed in this study seems to attain satisfactory registration quality as revealed from experimental results, which the authors claimed to outperform the conventional approaches in terms of registration accuracy and automation level.
The main idea of this study is to exploit the rich feature observations of building boundaries in urban scenes and acquire accurate 2-D line segments from optical image and 3-D line segments from LiDAR point cloud. Line matching between image space and object space is guided by matched point features and determined by line-point similarity invariant. Even though the authors implemented the approach in a way of employing point and line features, the line features finalize the object-to-image correspondence. It can be also interpreted as the line-based orientation where exterior orientation parameters are determined by control lines and line correspondence is automatically performed through the proposed steps. Both the derivation of accurate 3-D line feature from LiDAR and 2-D to 3-D line matching are all common tasks and have already been well studied. Thus, the authors’ work at its current form and content does not highlight much in terms of innovation. The following questions and suggestions may leave to the authors’ references for further improvement.
1. To refine the object-to-image correspondence, the corrections resulting from lens and film distortions are usually considered in the collinearity equations.
2. The quality of line segments both in image space and object space is crucial to the registration performance. The proposed method has not provided how the errors of observed line segments are evaluated or weighted in the transformation function.
3. It is not clear about how many and where the line-pairs are matched and utilized for refining the registration.
4. Are the nineteen line segments used for quantitative registration accuracy evaluation independent data set and where are they? And what are those A1 to A49 in Table 1?
5. Camera type used to acquire the experimental image has not been specified and whether camera calibration is placed has not been mentioned.
6. Line 281: “eave”->”eaves”
7. Line 332: What is set C?
8. Line 315: “Figure 3”->”Figure 4”
Author Response

(The authors gave the same response as above.)

Reviewer 3 Report
This paper suggest the use of point and line matching techniques for automatic registration between optical image and intensity image produced by LiDAR data. I would suggest this paper to be considered under major revision, many sections should be re-written.
In my opinion, this paper should be re-written in a careful manner, where methods should be more scientifically sound. Instead of describing your method, they could be transcribed mathematically.
There are several critical reasons why this paper is considered major revision.
1. I did not see how, in your introduction section your method can fill the research gap or out-performed the existing method.
2. You did not explain well why you are point matching first, and then line matching. Yes, you did mention that it’s for coarse registration, but there is no argument about why not directly apply line segment matching. Often by doing it twice you could introduce error, did you compared both ways and if so can you justify for it?
3. Please consider remove, or shorten “Transformation function based on the extended collinearity equations” it is basic knowledge that most reader who research in this topic should know.
4. Where is Figure 4? In your text you mentioned Figure 4 but I was not able to find it, please review your manuscript carefully before submitting it.
5. Figure and Fig should be use consistently
6. Why use Line Segment Detector, there are so many line detector out there could you justify your reasoning?
7. Your method heavily depend on line feature extraction results, but you didn’t show any results about it. What if the lines are overly generated? Or under generated? Are they accurate? How did you access your line feature extraction accuracy?
8. Why only use one image? One area? How does it work on another urban area?
9. Förstner, please fix accent all over the paper
10. Where is your discussion section? Error discussion?
11. Table 1 is very poorly formatted, please redo it.
12. You are not doing building detection, why only compare results derived from building edges?
13. One of your major conclusion is that “curves can serve as candidates that are able to mimic linear features. However, establishing a transformation function when curves are applied as registration primitives remains a technical problem worthy of further study”. I do not agree this this, here are tons of research for many years about image matching based on edge detection.
14. Please rewrite your discussion as well as conclusion.
Author Response

(The authors gave the same response as above.)

Round 2
Reviewer 2 Report
Most of the questions previously raised have been well addressed and treated, except that whether the 49 line segments are independent from the observed ones for estimating the registration, and how these check lines are distributed.
Author Response
First of all, thank you again for your careful reading and valuable comments. The 49 line segments for accuracy evaluation were selected and sketched manually. They are independent from the observed ones which were extracted by the methods described in the manuscript. The check lines were distributed evenly over the dataset but with various directions. We have added one sentence in the manuscript to describe the check lines (lines 390~392).

Reviewer 3 Report
I felt that this paper is still a weak paper, but after revision it is improved to acceptable. I hope in the future your manuscript can be carefully read through before submitting to journal. Thank you.
Author Response
Thank you so much for your pertinent suggestions! We had not been aware of our carelessness until you gave us so many valuable comments and suggestions based on your carefully reading. We deeply appreciate your contributions to the manuscript and we do realize how much it is important to have a careful reading before a manuscript is submitted to a journal. Thank you again for your great reviewing works!